# Effect of CrossFit Training on Physical Fitness of Kickboxers

**DOI:** 10.3390/ijerph19084526

**Published:** 2022-04-08

**Authors:** Tadeusz Ambroży, Łukasz Rydzik, Amadeusz Kwiatkowski, Michał Spieszny, Dorota Ambroży, Aneta Rejman, Agnieszka Koteja, Jarosław Jaszczur-Nowicki, Henryk Duda, Wojciech Czarny

**Affiliations:** 1Institute of Sports Sciences, the University of Physical Education in Krakow, 31-541 Kraków, Poland; uksgladiator@gmail.com (A.K.); michal.spieszny@awf.krakow.pl (M.S.); dorota.ambrozy@awf.krakow.pl (D.A.); agnieszka.koteja@gmail.com (A.K.); henryk.duda@awf.krakow.pl (H.D.); 2College of Medical Sciences, Institute of Physical Culture Studies, University of Rzeszow, 35-310 Rzeszów, Poland; anetarejman13@wp.pl (A.R.); wojciechczarny@wp.pl (W.C.); 3Department of Tourism, Recreation and Ecology, University of Warmia and Mazury in Olsztyn, 10-719 Olsztyn, Poland; j.jaszczur-nowicki@uwm.edu.pl; 4Department of Sports Kinanthropology, Faculty of Sports, University of Presov, 080 01 Presov, Slovakia

**Keywords:** kickboxing, CrossFit, training effectiveness, general fitness

## Abstract

Background: Kickboxing is a combat sport that is complex in technique, tactics, and movement structure, and requires an adequate level of motor skills as a foundation for activities during competitions. General physical fitness, defined as the effect of the externalization of motor skills, is the basis for athletic training regardless of the sport. The aim of this study was to determine the effect of modified training based on the principles of CrossFit on the development of general physical fitness in a group of kickboxers compared to a control group. Methods: The study was experimental in nature and was conducted in a group of 60 kickboxers, divided into experimental and control groups. Participants were selected by purposive sampling, and the criteria were training experience, sports skill level (minimum class 1 athletes), and consent to participate in the experiment. The intervention in the study group involved the introduction of CrossFit-based training into a conventional kickboxing training program. General and special physical fitness of the athletes were diagnosed. Results: Statistically significant differences were found in general fitness in terms of abdominal strength (*p* < 0.001), pull-ups (*p* < 0.001), dynamometric measurement of handgrip force (*p* < 0.001) (kg), clap push-ups (*p* < 0.001), standing long jump (*p* < 0.001), shuttle run (*p* < 0.001), sit-and-reach (*p* < 0.001), and tapping (*p* < 0.001). Furthermore, changes in special fitness were also demonstrated for the special kickboxing fitness test (SKFT) (*p* < 0.02), the total number of punches (*p* < 0.001), punching speed (*p* < 0.001), and hip turning speed (*p* < 0.001). There was also a correlation between characteristics of general fitness and special fitness (*p* < 0.001). Conclusions: The experimental training program based on the principles of CrossFit training had a positive effect on the general and special kickboxing physical fitness.

## 1. Introduction

General physical fitness, defined as the effect of the externalization of motor skills, is the basis for athletic training regardless of the sport [1,2,3]. By properly developing its level, it is possible to implement more precise and effective specialized training forms, which are a prerequisite for athletes’ performance during tournaments [4,5,6]. Research also shows that inadequate levels of fitness may increase the likelihood of injury [7]. The development of the athlete’s motor skills depends on sport [1]. Kickboxing is a combat sport that is complex in technique, tactics, and movement structure, and requires an adequate level of motor skills as a foundation for activities during competitions [8,9,10].

Research on the effectiveness of combat sports training indicates that it is highly dependent on general motor skills [11,12,13,14]. For kickboxers, the basis for high competitive performance is a high level of strength, dynamics, anaerobic capacity, and aerobic capacity, with the latter considered the necessary foundation for all activities. Therefore, physical preparation should be based on improving upper limb strength and speed (punches and combinations during offensive actions and blocks and dodges during defensive actions) and developing high levels of anaerobic power (dynamic kicks and punches) [15,16].

Training efficiency, which is the ability to endure loads based on sufficient physical capacity, is also an important element that increases the competitive performance of athletes [17,18,19,20]. The specifics of training at the elite level should be tailored to the individual athlete’s needs. In search of opportunities to improve work efficiency during training, modern forms of training are implemented and verified, often differing from the classic training methods commonly used in kickboxing, which include repetition methods (at maximal or submaximal work intensity, full rests) and interval methods (low, medium, or high intensity, and incomplete rests) based on technical elements using special coaching devices such as focus mitts, kick shields, punching bags, etc. [5,21].

The growing popularity of CrossFit is of interest to researchers in the context of implementing this training modality, with its hallmark being competing during exercise [22,23]. This type of training, based on exercises derived from the arsenal of training means used in gymnastics, weightlifting, and strength and functional training, is focused on the continuous improvement of the athlete’s performance in individual exercises through competition with a training partner or themself [24]. This model of strength and conditioning training can also be applied in the kickboxing environment.

The implementation of new forms of training and the detailed verification of their effects on the physical capabilities of athletes are widespread in combat sports [25,26,27,28,29]. Through the implementation of experimental training plans, it becomes possible to improve the quality of training, which can, in turn, affect the competitive performance of athletes [30]. The implementation of modern experimental forms of training in kickboxing is popular, and the novelty of our study is the introduction of the author’s form of training based on CrossFit training with exercises commonly used in kickboxing. Contemporary analyses in the context of the implementation of CrossFit training in the structure of kickboxing include the effect of CrossFit exercises on the body posture of kickboxers [31], and a comparison of CrossFit training and other forms of resistance training to maintain an optimal level of fitness [32].

To determine the athlete’s level of preparation for the competition, an analysis of a fight is performed [16,33,34] by determining the technical and tactical indices or by evaluating special physical fitness using tests corresponding to the structure of a kickboxing bout [35]. This type of analysis was conducted in the present study.

The aim of this study was to determine the effect of the author’s training according to the principles of CrossFit on the development of general and special physical fitness in a group of kickboxers compared to a control group. The choice of general fitness tests was based on examinations performed in previous studies [16,33]. In this study, an attempt was made to find if the proposed experimental training program could be successfully implemented into the regular strength and conditioning programs in kickboxing.

## 2. Material and Methods

### 2.1. Study Group

The study was experimental in nature and was conducted in a group of 60 male kickboxers. Based on the sample size, as determined using G*Power software 3.1.9.4, it was shown that, with the estimated moderate effect size, at least twenty participants were needed in each group (effect size f = 0.65, power = 0.95, *p* = 0.05) [36]. In the present analysis, 30 participants in each group were recruited to ensure a more accurate analysis. Participants were selected by purposive sampling, i.e., subjective non-random sampling based on clear criteria for selecting a group for the study such as training experience, sports skill level (minimum class 1 athletes; in Poland, athletes are classified by sports skill classes: there are classes 1, 2, and 3, with each class determined based on winning during competitions), consent to participate in the experiment, and good health status. The inclusion criteria adopted were due to the high intensity of the proposed form of training that requires the participants to have bodies adapted to such a high level of exercise. The selected group was randomly divided into two (experimental and control) groups, each consisting of 30 athletes.

In the control group, training was conducted according to a standard general cycle. In the study group (experimental), training was modified (independent variable) by introducing an experimental program (CrossFit workout) into their standard training. The experiment required a high degree of accuracy and maximum commitment from the athletes. During the experiment, the athletes did not participate in sporting events or sparring at competitive intensity levels and did not report any injuries. Observation of the changes in the experiment and their quantitative evaluation were also performed. The dependent variable in this case was general and special physical fitness, viewed as differences in the fitness test results in both groups.

### 2.2. Morphological Characteristics of the Subjects

The mean training experience in all participants was 8.1 ± 4.24 years. They trained 1.5 to 2 h, 6 to 8 times per week. The mean age of study participants was 20.07 ± 1.46 years, body weight was 73.56 ± 8.13 kg, body height was 179.55 ± 0.45 cm, body fat was 14 ± 0.2%, and BMI was 23.63 ± 1.19 kg/m^2^. Body composition and body weight were measured using a Tanita BC-601 body composition monitor (Tanita, Tokyo, Japan) [37] in the morning in fasting conditions, whereas body height was measured using a SECA 2017 body height meter (Seca, Hamburg, Deutschland). The examination was performed in the standing position, as recommended by the manufacturer. The subjects did not consume alcohol the day before the measurement and avoided large meals and extreme physical exercise.

### 2.3. Research Program and Methodology

According to the principles of the implementation of pedagogical experiments, the researcher’s interference involved manipulation of the training process in the experimental group. The first examination took place before the experimental training in both groups, in the control and preparation mesocycle of the preparation period. Next, the group of participants performed a modified workout (by introducing an experimental program based on CrossFit workout into their standard training programs) three times a week during each training session. During this time, the control group performed standard kickboxing training without modification. After eight weeks, another examination was conducted (effect control).

Furthermore, each participant was instructed not to use specialized diets and supplementation during the experiment due to the strong relationship between diets and results obtained during tests. Diets were monitored using notebooks in which the participants recorded the foods they consumed using home measures based on a photo album of foods and products. The recording procedure was continued for 3 days: 2 working days and 1 day off [38]. The analysis of diet observations revealed neither specific diets nor the use of performance-enhancing supplements in the training groups. The experiment was approved (No./309/KBL/OIL/2019) by the Bioethics Committee of the Regional Medical Chamber in Kraków, Poland.

### 2.4. Principles of the Experimental Training

Using the principles of CrossFit training methodology [39] and typical kickboxing training exercises based on the most frequently performed techniques [40], a training program was developed to improve the physical fitness of kickboxers (Table 1).

The experimental training program is based on the AMRAP method (Table 2), which consists of performing as many sets of efforts as possible with a fixed number of repetitions in a given time (10 min in this case) [39]. To avoid boredom and training routine, a different training unit was planned for each workout of the week. Each workout was preceded by a warm-up and consisted of 8 exercises including boxing punches, kickboxing kicks, and exercises typical of the CrossFit methodology such as burpees (transition from standing upright to squat position, kicking the feet back to a front support position, returning to squat position, and jumping up), or box jumps. The training program was designed so that it was simple to perform and accessible to every participant. Therefore, additional equipment was kept to a minimum.

### 2.5. Physical Fitness Tests

Physical fitness was evaluated using selected items of the International Committee on the Standardization of Physical Fitness Test (ICSPFT) [2] and Eurofit Physical Fitness Test (EUROFIT) [41]. The entire test battery consisted of the following tests:Cooper test (in m). A running endurance test consisting of 12 min of uninterrupted running. The running distance is measured [42].Dynamic sit-ups (in reps). Evaluation of abdominal strength: the tested person lies on the mattress with feet 30 cm apart and knees bent at a right angle. Hands are intertwined, resting on the neck. The participant is assisted by a partner who holds the participant’s feet so that they remain in contact with the ground. At the start signal, the participant sits up to touch the knees with elbows and then returns to the starting position. The exercise duration is 30 s [41].Pull-ups (in reps). Evaluation of the strength of the shoulder girdle using the number of repetitions: the participant catches the bar with a pronated grip and hangs; at the signal, the participant bends their arms at the elbows and pulls the body up so high that the chin is above the bar and then, without a rest, returns to a simple hanging; the exercise is repeated as many times as possible without a rest; the result is the number of complete pull-ups (chin over the bar) [2].Measuring handgrip strength with a dynamometer (evaluation of static force). The participant stands with a small straddle with a dynamometer held tightly in the fingers. The arm is positioned along the body so that the hand does not touch the body, and the participant performs a short grip on the dynamometer with maximum force, with the other arm resting along the body. The better result of the two tests of maximal static strength (in kgf) of the dominant hand (HGSmax) using a handgrip dynamometer (MG 4800, Charder, Taichung, Taiwan) was recorded, with an accuracy of 1 kg. The better score of the right and left hand tests was recorded, and the interval between the tests was 5 min [41].Flexibility test: sit-and-reach (in cm). The test is performed as a sit-and-reach movement, with the range of motion measured in cm, below the feet level. In a seated position, the participant reaches their arms forward as far as they can. The participant, in a straddle sitting position, reaches forward with the hands as far as possible by sliding the ruler on the surface of the box with a previously prepared scale. The better of the two results is recorded. If the participant reaches 10 cm beyond the toes, they achieve a score of 10. A box 40 cm long, 45 cm wide, and 35 cm high, a 65 cm long graduated box top protrudes 25 cm over the side wall that marks the width of the box and is used as a feet rest; the box top is fixed in such a way that the graduation mark drawn on it indicates 50 in the place where feet touch the surface of the box. A 30 cm-long ruler is placed loosely on the surface of the box perpendicularly to its longitudinal axis and used for moving with hands while performing a forward reach [41].Shuttle run (in s). The participant runs on a signal to the second line 5 m away, crosses it with both feet, and comes back. They run 10 times for a distance of 5 m; the time of the shuttle run is measured and rounded to a decimal place of a second [41].Tapping (assessment of the speed of upper limb movement). The participant stands with a small straddle, putting the non-dominant hand on a rectangular plate; the dominant hand should be placed crosswise on the opposite plate, and then the participant touches both plates alternately as quickly as possible. The participant performs a total of 50 movements, i.e., touches each plate 25 times. The better of the two results is recorded, determined by the time it takes to touch each plate 25 times, measured to the nearest 0.1 s. The equipment needed includes an adjustable height table (or vaulting box), two rubber discs 20 cm in diameter horizontally attached to the table with their centers 80 cm apart, a 10 *×* 20 cm rectangular plate placed in the middle between them, and a timer [41].Clap push-ups (n)—in the front support position with feet on a gym bench (30 cm), the participant performs the maximum number of push-ups from the ground with a hand clap [43].Standing long jump (in cm). The participant stands with the feet slightly apart in front of the starting line, bends the knees, and moves the arms backward at the same time, and then they perform an arm swing and jump as far as they can; the landing occurs on both feet while maintaining the upright position; the test is performed twice. The longest of the two jumps measured to the closest mark left by the participant’s heel is recorded, with an accuracy of 1 cm [41].To assess special fitness, the special kickboxing fitness test (SKFT) [35] and batteries of special fitness tests designed for combat sports in the standing position were used [44].To evaluate special fitness levels and technical skills, all participants underwent the special kickboxing fitness test (SKFT) [45]. Description of the procedure for special kickboxing physical fitness test: Prior to performing the test, participants performed a warm-up that included 5 min of an easy run and 10 min of general warm-up and stretching (flexibility) exercises. The following tools were prepared to perform the test: adhesive tape to mark distances on the mat, a stopwatch to measure time, kick shields and punch shields, a protocol for recording the results, and a sport tester (heart rate monitor). In the first station, the athlete performs, from a fighting stance, a combination of punches to the shield held by the partner: left and right straight punches to the head, without stopping, for 30 s. After completing this part of the test, the athlete runs 10 m in a straight line to the next station (No. 2), where, from the fighting position, they perform roundhouse kicks to the shield held by the partner for 30 s: left high kick (high roundhouse kick) and right high kick to the head. Next, the athlete runs back to the first station with shields and performs a left straight-right hook combination for another 30 s to the head. After completion of this part of the test, the athlete runs 10 m to the partner holding the shield in station 2 and performs middle roundhouse kicks for 30 s alternately with the right and left leg to the body trunk. The total special exercise time during the test is 2 min (4 × 30 s). Correctly performed kicks and punches were counted in each of the four parts. Heart rate (HR, bpm) was measured directly after completion of the test and after 1 min rest. The Garmin HRM chest strap was used in the tests. The proposed special fitness test allowed for the evaluation of the technical level of athletes in terms of the most effective and most frequently used hand techniques (punches) and leg techniques (kicks), speed (number of punches and kicks performed per time unit), special endurance (response of the circulatory system and number of punches and kicks), coordination (combination of kicks and punches), and flexibility (kicking range). The 10 m running distance used to move between stations corresponds to the diagonal of the largest ring found in ring combat sports. The technical skills used in the test ensure the selectivity of the test, making it inaccessible to those who do not perform special training and do not have the appropriate level of technical proficiency. Furthermore, after the test, based on the results obtained, the index of special fitness was calculated using a specialized formula:
$$\frac{\mathrm{Final}\;\mathrm{HR}\left(\mathrm{bpm}\right)+\mathrm{HR}1\min\left(\mathrm{bpm}\right)}{\mathrm{Kick}+\mathrm{Punches}\;\left(N\right)}$$
where:


Final HR—heart rate recorded immediately after completion of the test;HR1 min—heart rate recorded 1 min after completion of the test;Kicks—the number of kicks performed in the test;Punches—the number of punches performed in the test.


The special fitness index reflects the level of a fighter’s special fitness, which means the effective interaction of the body’s exercise capacity, general fitness, and the athlete’s technical skills. The interpretation of the score is inversely proportional: the higher the level of special fitness, the lower the value of the kickboxing test index.

12.Speed punches test. The punches are performed from a fighting stance. Each participant performs a combination composed of two punches: a left straight punch (Jab) to the head and a right straight punch (Punch) to the body trunk without changing the distance. The shields to which the participant performs 30 such combinations (60 punches in total) are held by a partner at the constant height. The time needed to perform 30 complete combinations is recorded in seconds to the nearest 0.1 s.13.Hip-turning speed test: in the hip-turning speed (frequency) test, each athlete has a belt attached over the right hip (unless they fight in the opposite position) and, using the fighting stance, turns their hips to the left. This movement causes tension of the belt held by the coach standing behind the athlete (control). Next, the participant returns the hip. The participant is instructed to perform 30 hip turns (the number of belt tension instances is counted). The time taken to perform 30 turns was recorded.

The tests were conducted in a three-day mode at noon: trials 1–5 on the first day, trials 6–9 on the second day, and special fitness tests on the third. All tests were performed before and after the experiment, except for tapping, standing long jump, and flexibility tests, which were performed twice, and the better score was recorded. The intervals between the tests were designed so that the participants rested completely. A 20 min warm-up was conducted with the athletes prior to testing.

### 2.6. Statistical Analysis

Statistical analysis of logarithmic data was performed using STATISTICA v13.1 PL software (Statsoft, Kraków, Poland). Basic descriptive statistics were computed: arithmetic means, standard deviations, and 95% confidence intervals. The data were tested for normal distribution using the Shapiro–Wilk test, and Student’s *t*-test for dependent variables was used to assess the significance of differences. Student’s *t*-test for independent variables was used to determine the significance of differences between the control and experimental groups. The effect size was calculated using Cohen’s d index. When d ranges from 0 to 0.2, the effect is small, i.e., negligible; it is medium from 0.2 to 0.5, large from 0.5 to 0.8, and extremely large when over 1.4. The relationships between the data were verified using Pearson linear correlation. Correlation values were interpreted as weak for the ranges of −0.5 to 0.0 or 0.0 to 0.5, and strong for −1.0 to −0.5 or 0.5 to 1.0 [46,47]. The level of statistically significant differences was set at *p* < 0.05.

## 3. Results

Statistically significant differences were found in the results of measurements before and after the experiment in the experimental group. Significant improvements in performance were noted in all tests except for the Cooper test. The analysis of the results (Table 3) indicated a statistically significant increase in abdominal muscle strength (5% difference), handgrip strength, an increase in the number of pull-ups (12% difference), an increase in the number of clap push-ups (9% increase), and an increase in the length of the long jump (1.8% difference), as well as a decrease in the time of the shuttle run (3% difference) and an increase in the results of the sit-and-reach test (8% difference) compared to the results of the initial test (pre-test compared to post-test). No statistically significant differences in the results of the Cooper test were observed (*p* > 0.05). In the control group, the only significantly improved result was the static strength test (handgrip test on a dynamometer). A comparison of the difference between the groups after the experiment revealed statistically significant differences in abdominal strength, clap push-ups, standing long jump, sit-and-reach, and tapping (Table 3).

The analysis of the changes in special fitness induced by the experiment showed a significant improvement in all the parameters measured in the experimental group. Furthermore, there were significant adverse changes in the control group in punch speed and hip-turning speed. The difference between the groups after completion of the experiment on the special kickboxing fitness test (SKFT) and the total number of punches in the test and the hip-turning speed were also found to be statistically significant (Table 4).

There were statistically significant relationships of endurance as measured by the Cooper test, total number of punches in the SKFT, speed of upper limb movements (tapping) with punching speed, and agility run with hip-turning speed. All the correlations were high and statistically significant at *p* < 0.001 (Table 5).

## 4. Discussion

In this study, we attempted to determine the effect of CrossFit training on the general and special physical fitness of kickboxers. The results presented in this study allow for the verification of an experimental program based on CrossFit training. The idea of CrossFit training is to introduce a component of competing with a training partner or oneself, which increases the intensity and effectiveness of exercise [24]. The findings of the study show that the experiment had a positive effect on the fitness level in the experimental group, which was confirmed by statistically significant changes. Improvements were observed in abdominal muscle strength, shoulder girdle strength, and handgrip strength. The development of general body strength is a basic component of the preparation of a kickboxer to effectively use hand and foot techniques and to improve performance during the fight [4,48,49]. Furthermore, strength training is effective in protecting against injuries that are common in contact sports [50,51]. The results of our research show that it is advisable to use CrossFit training as a supplement to basic kickboxing training in the preparation period. The experimental training also improved dynamic upper and lower limb strength as measured by standing long jump and clap push-ups. Muscle strength of both upper and lower limbs is essential for the athlete to win in full-contact formulas in kickboxing such as K1, low kick, and full contact [52,53]. Taking into account the significant changes, it can be concluded that the training used in the experiment led to the increase in dynamic strength in the experimental group despite the lack of exercises with additional external resistance. A high level of dynamic power allows for the effective use of technical actions that affect the level of technical and tactical indices and thus fighting performance [16,33]. Another effect of the implemented training program was a progression in speed, as confirmed by statistically significant changes recorded in the test of speed of hand movements (plate tapping test) in the experimental group. Accordingly, the authors’ program induced improvements in upper limb speed in plate tapping and special fitness tests (punching speed). The relationship between these two components was confirmed by the high correlation between each other. It is worth noting that these abilities play a key role in full-contact formula (K1, low kick, full contact), light forms (kick light, light contact), and intermittent forms (point fighting) [54], as well as in any fight using upper limbs (boxing, karate, etc.). According to Kimm and Theil, hand speed is especially important in boxing, both to protect against attack and throw punches [55]. A kickboxer who delivers punches at a fast pace can effectively attack the opponent, anticipate their intentions, and effectively defend against the opponent’s attack. A high frequency of upper limb movements and punches is highly desirable, especially in limited-contact formulas where a point advantage determines victory [9]. The observed progression can be explained by the use of plyometrics in the experimental training involving the upper limbs, such as burpees, combined with dynamic, explosive boxing actions on a bag. The purpose of these actions was to develop the muscles’ ability to generate a large force in a short time (high force rate) or improve dynamics [56]. In addition to these characteristics, agility, balance, and coordination also play an important role in the training process [15]. An improvement in the results after the experimental training was also observed in the agility test (shuttle run). This ability determines the speed of movement, which is a precondition for effective movement in the fighting area and defense. Furthermore, the ability to move quickly and the footwork speed are the basis for meeting the technical and tactical objectives of the fight [57].

In the present study, a high correlation was observed in the experimental group between the hip-turning speed in the special fitness test and shuttle run results. Interpretation of the relationships leads to the conclusion that the kick speed, associated with hip turning, is determined by agility, with its components being coordination and speed [2,15]. There was a significant increase in flexibility in the experimental group after the experimental training program. Flexibility is one of the elements needed to master technical skills [15]. The sit-and-reach test showed the progression of flexibility in athletes in the study group. The improvements may be due to the training used, which included high kicks that required a large range of motion. The Cooper Test, which measures the athlete’s aerobic endurance level, showed small and statistically insignificant improvements in both groups. This may be explained by high baseline levels of running endurance in the kickboxers tested, whereas the specific kickboxing training and experimental training did not allow for significant development of this ability. It is worth noting that the higher level of the Cooper test results translates directly into higher special fitness, as evidenced by the high positive correlation with the total number of punches in the test. Furthermore, the experimental group showed a statistically significant reduction in the physical fitness index in the SKFT test, which indicates an improvement in the athletes’ special endurance. Favorable changes were found in all special fitness parameters, i.e., the total number of punches, punching speed, and hip-turning speed. This leads us to assume that the applied training based on the components of CrossFit has a positive impact on the athlete’s performance, which is measured by the results of special fitness tests [45,58].

### Limitation in the Study

A major limitation of our study was the lack of detailed verification of the study groups. We were unable to conduct the experiment in a closed facility where both groups would have the same conditions for functioning and training. Furthermore, we did not perform a direct test to evaluate VO_2_max based only on running endurance.

## 5. Conclusions

The experimental training program based on the principles of CrossFit training had a positive effect on the physical fitness of the kickboxers in terms of strength, flexibility, agility, and speed indices in the experimental group. The experimental training program based on the principles of CrossFit training had a positive effect on special physical fitness. There were significant correlations in the parameters of special and general fitness, i.e., Cooper test vs. total of punches and kicks, shuttle run vs. hip-turning speed, and tapping vs. punching speed after the experimental training program.

### Practical Implication

The presented experimental training program can be implemented to improve the quality of kickboxing training in terms of improving general and special physical fitness in the preparation period.

## Figures and Tables

**Table 1 ijerph-19-04526-t001:** Methodology of the experimental training program.

Experimental Training Program: Methodology	
Number of exercises	8
Method and duration	AMRAP 10 min
External resistance	Body weight
Exercise intensity	Submaximal
Rests	No rest

AMRAP—As many rounds as possible, a method of training that involves performing as many sets of efforts as possible at a fixed number of repetitions in a given time.

**Table 2 ijerph-19-04526-t002:** The experimental training program used in the study.

Workout 1: Monday	Workout 2: Wednesday	Workout 3: Friday
- 40 × punching bag techniques: jab, punch - 40 × air squat - 30 × kick pad techniques: left. middle roundhouse kick 2×, right. middle roundhouse kick - 30 × box jumps (40 cm) - 20 × punching bag techniques: jab, punch, left middle roundhouse kick - 20 × sit-ups - 10 × burpees + punching bag combinations: jab, punch, high roundhouse kick - 10 × push-ups	- 60 × mountain climbers - 10 × punching bag techniques: jab, punch, left middle roundhouse kick - 30 × Russian twists - 30 × punching bag techniques: jab, punch, left. middle roundhouse kick - 30 × lunges - 15 × air squat + left middle front kick/right middle front kick - 15 × box jumps - 10 × burpees + punching bag combinations: left middle roundhouse kick 2×/right middle roundhouse kick 2×	- 50 × single under - 20 × (push-ups 2× + punching bag techniques: jab, punch) - 20 × spinal rock - 20 × (air squat 2× + left middle roundhouse kick/right middle roundhouse kick - 10 × hand release push-ups - 10 × burpees + kick pad combinations: jab, punch - 10 × tuck jumps - 5 × combinations of any 5 kicks on the bag

A detailed video illustration of the exercises is provided in the Appendix A.

**Table 3 ijerph-19-04526-t003:** General physical fitness of the participants before and after the experiment.

Parameters	Pre-Test		Post-Test		Mean Difference (95%CI) Pre-Post Test				Student’s *t*-Test		Effect Size
	Mean	SD	Mean	SD	Mean	SD	−95%CI	+95%CI	*t*	*p*	Cohen’s D
**Abdominal strength (n) E**	25.97	3.79	27.47	3.42	−1.5	2.1	−2.28	−0.71	−**3.91**	<**0.001**	−0.11
**Abdominal strength (n) C**	24.66	3.20	24.40	3.10	0.26	0.10	−0.12	0.66	1.39	0.17	0.08
**Between group**	t = 1.43 *p* = 0.15 d = 0.37		**t** = **3.63** ***p*** = <**0.001** d = 0.94								
**Pull-ups (n) E**	6.73	3.12	7.60	3.39	−0.87	1.25	−1.33	−0.40	−**3.79**	<**0.001**	−0.80
**Pull-ups (n C**	7.20	3.19	7.43	4.03	−0.23	−0.84	−0.99	0.52	−0.62	0.53	0.06
**Between group**	t = −0.57 *p* = 0.56 d = 0.15		t = 0.17 *p* = 0.86 d = 0.05								
**Dynamometric measurement of handgrip force (kg) E**	51.99	2.72	52.51	2.79	−0.52	0.40	−0.66	−0.37	−**7.14**	<**0.001**	−0.07
**Dynamometric measurement of handgrip force (kg) C**	52.11	3.01	52.69	3.13	−0.58	−0.12	−0.92	−0.22	−**3.29**	**0.02**	0.19
**Between group**	t = −0.16 *p* = 0.86 d = 0.04		t = −0.22 *p* = 0.82 d = 0.06								
**Clap push-ups (n) E**	11.03	4.87	12.23	5.53	−1.2	1.28	−1.67	−0.73	−**5.17**	<**0.001**	−0.04
**Clap push-ups (n) C**	11.66	5.03	11.43	4.73	0.23	0.3	−0.18	0.65	1.15	0.25	0.05
**Between group**	t = −0.49 *p* = 0.62 d = 0.13		**t** = **0.60** ***p*** = **0.04** d = 0.16								
**Standing long jump (cm) E**	201.2	14.9	204.83	15.66	−3.63	2.77	−4.67	−2.60	−**7.17**	<**0.001**	−0.02
**Standing long jump (cm) C**	201.3	11.97	201.46	12.37	−0.16	−0.4	−1.09	0.76	−0.36	0.17	0.01
**Between group**	t = −0.02 *p* = 0.97 d = 0.01		**t = 0.92*****p*** = **0.03** d = 0.24								
**Shuttle run (s) E**	19.37	2.58	18.71	2.23	0.63	0.74	0.36	0.91	**4.68**	<**0.001**	0.11
**Shuttle run (s) C**	18.61	2.08	18.64	2.07	−0.03	0.01	−0.07	0.00	−2.04	0.06	0.01
**Between group**	t = 1.19 *p* = 0.23 d = 0.33		t = 0.10 *p* = 0.92 d = 0.03								
**The Cooper test (m) E**	2375.83	240.75	2390.33	213.25	−14.5	237.87	−103.32	74.32	−0.33	0.74	0.00
**The Cooper test (m) C**	2400.60	213.37	2398.40	228.94	2.2	−15.57	−42.16	18.56	−0.79	0.43	0.01
**Between group**	t = −0.91 *p* = 0.36 d = 0.11		t = −0.89 *p* = 0.37 d = 0.04								
**Sit-and-reach (cm) E**	10.33	4.67	11.27	4.75	−0.93	1.34	−1.43	−0.43	−**3.82**	<**0.001**	−0.04
**Sit-and-reach (cm) C**	11.26	3.77	11.13	3.73	0.13	0.04	−0.19	0.45	0.84	0.40	0.03
**Between group**	t = −0.85 *p* = 0.39 d = 0.22		**t** = **3.37** ***p*** = <**0.001** d = 0.03								
**Tapping E**	12.41	1.48	11.76	1.36	−0.65	0.52	−0.46	−0.84	**6.92**	<**0.001**	0.41
**Tapping C**	12.09	1.38	11.98	1.45	0.11	0.07	−0.62	1.01	0.49	0.62	0.08
**Between group**	t = 0.84 *p* = 0.40 d = 0.22		**t** = **1.14** ***p*** = **0.05** d = 0.16								

E—experimental group, C—control group, SD—standard deviation, CI—confidence intervals, statistically significant values are bolded.

**Table 4 ijerph-19-04526-t004:** Special physical fitness of the participants before and after the experiment.

Parameters	Pre-Test		Post-Test		Mean Difference (95%CI) Pre-Post Test				Student’s *t*-Test		Effect Size
	Mean	SD	Mean	SD	Mean	SD	−95%CI	+95%CI	*t*	*p*	Cohen’s D
**SKFT- Index E**	191.13	9.07	187.80	9.32	3.32	0	0.57	6.09	**2.47**	**0.02**	0.46
**SKFT- Index C**	189.70	9.87	188.51	10.48	1.92	−0.61	−2.94	6.74	0.80	0.42	0.12
**Between group**	t = 0.58 *p* = 0.56 d = 0.15		**t = 1.24 *p* = 0.05 d = 0.43**								
**Total number of punches E**	280.13	30.81	293.26	30.76	−13.3	0.05	−15.73	−10.53	**−10.32**	**<0.001**	0.43
**Total number of punches C**	286.15	29.63	278.26	33.17	7.86	−3.54	−2.23	17.96	1.59	0.12	0.25
**Between group**	t = -0.76 *p* = 0.44 d = 0.20		**t = 1.72 *p* = 0.04** d = 0.47								
**Punching speed E**	16.27	1.34	15.76	1.32	0.37	0.02	−0.38	0.63	**8.52**	**<0.001**	0.38
**Punching speed C**	16.15	1.39	16.22	−1.42	0.07	−0.07	−0.14	−0.00	**−2.12**	**0.04**	0.05
**Between group**	t = −0.33 *p* = 0.74 d = 0.09		t = −1.29 *p* = 0.20 d = 0.34								
**Hip turning speed E**	16.40	1.58	15.89	1.63	0.51	−0.05	0.32	0.71	**5.35**	**<0.001**	0.32
**Hip turning speed C**	16.64	1.42	16.82	1.35	−0.18	0.07	−0.30	−0.06	**−3.04**	**0.03**	0.13
**Between group**	t = −0.65 *p* = 0.50 d = 0.16		**t = −2.40 *p* = 0.02** d = 0.62								

E—experimental group, C—control group, SD—standard deviation, CI—confidence intervals, statistically significant values are bolded.

**Table 5 ijerph-19-04526-t005:** Relationships between special physical fitness and general physical fitness of kickboxers after the experimental program.

Pearson Linear Correlation Coefficient	SKFT-Index	Total Number of Punches	Punching Speed	Hip Turning Speed
Abdominal strength	−0.67	−0.57	−0.09	0.11
	*p* > 0.05	*p* > 0.05	*p* > 0.05	*p* > 0.05
Pull-ups	0.01	0.08	0.22	−0.06
	*p* > 0.05	*p* > 0.05	*p* > 0.05	*p* > 0.05
Clap push-ups	0.01	0.10	0.08	−0.13
	*p* > 0.05	*p* > 0.05	*p* > 0.05	*p* > 0.05
Standing long jump	−0.01	0.33	−0.27	0.16
	*p* > 0.05	*p* > 0.05	*p* > 0.05	*p* > 0.05
Shuttle run	−0.30	0.05	0.24	**0.81**
	*p* > 0.05	*p* > 0.05	*p* > 0.05	***p* < 0.001**
The Cooper test	0.13	**0.76**	−0.28	−0.24
	*p* > 0.05	***p* < 0.001**	*p* > 0.05	*p* > 0.05
Sit-and-reach	0.03	−0.23	−0.29	−0.07
	*p* > 0.05	*p* > 0.05	*p* > 0.05	*p* > 0.05
Tapping	0.19	0.18	**0.84**	−0.08
	*p* > 0.05	*p* > 0.05	***p* < 0.001**	*p* > 0.05

Statistically significant values are bolded.

## Data Availability

The data presented in this study are available on request from the corresponding author.

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
