# Peer review of "Effect of CrossFit Training on Physical Fitness of Kickboxers"

_ijerph, 2022, doi:10.3390/ijerph19084526_

Round 1

Reviewer 1 Report

Thank you for the opportunity to review the paper entitled " Effect of CrossFit training on physical fitness of kickboxers".

The proposal for contact sports such as kickboxing is interesting. However, the authors must resolve several methodological issues:

How to know if the increase in physical fitness also increases sports performance in kickbonxing? According to the exposed antecedents, this question has no answer. This is because the authors did not use a kickboxing-specific test.

The authors did not use a control group to compare results, nor did they use a specific test to observe kickboxing performance.

L52-54: what is the arsenal of training methods commonly used in kickboxing?

L123: Review Reference 2

L189-194: Include parameters for Cohen's d and Pearson correlation

L208-119: it is logical that training to improve physical fitness improves physical fitness. But which is the comparison? which is the control group? which is the second experimental group? which is the test that defines the increase in sports performance in kickboxing?

L220-221: Why are anthropometric variables correlated with fitness tests? is this a way to demonstrate performance in kickboxing?

Also, neither correlation was high. Does this mean that the sports level of the athletes is low? or what does it mean?

L222-271: Only the results are described in the discussion. It is not discussed with other investigations that have used similar methodologies

L276-278: Qué sugerencias entregarían a entrenadores de kickboxing?

Author Response

Dear Reviewer,

Thank you very much for your time and valuable comments, which all have been considered and incorporated. The detailed list of responses is given below. We hope that the modifications and explanation will be acceptable for you.

Yours sincerely,

Rydzik, corresponding author

Thank you for the opportunity to review the paper entitled " Effect of CrossFit training on physical fitness of kickboxers".

The proposal for contact sports such as kickboxing is interesting. However, the authors must resolve several methodological issues:

How to know if the increase in physical fitness also increases sports performance in kickbonxing? According to the exposed antecedents, this question has no answer. This is because the authors did not use a kickboxing-specific test.

A: Thank you very much, we have the results of the special kickboxing test and we have included them in the study. We have not done this so far because we wanted to write a separate paper on special fitness.

The authors did not use a control group to compare results, nor did they use a specific test to observe kickboxing performance.

A: The results of the control group have been included. As previously, we have not included them before to ensure the clarity of statistical analysis.

L52-54: what is the arsenal of training methods commonly used in kickboxing?

A: This part has been corrected

L123: Review Reference 2

A: We are very sorry but we do not understand this comment, could you please be more specific?

L189-194: Include parameters for Cohen's d and Pearson correlation

A: We have removed the correlations and added information on the effect size.

L208-119: it is logical that training to improve physical fitness improves physical fitness. But which is the comparison? which is the control group? which is the second experimental group? which is the test that defines the increase in sports performance in kickboxing?

A: The control group has been added and the manuscript has been rewritten.

L220-221: Why are anthropometric variables correlated with fitness tests? is this a way to demonstrate performance in kickboxing?

A: Thank you for your comment, correlation has been changed to include special fitness in the manuscript.

Also, neither correlation was high. Does this mean that the sports level of the athletes is low? or what does it mean?

A: The correlation has been corrected

L222-271: Only the results are described in the discussion. It is not discussed with other investigations that have used similar methodologies

A: The discussion has been corrected

L276-278: Qué sugerencias entregarían a entrenadores de kickboxing?

A: This part has been corrected

Reviewer 2 Report

Dear Authors

You have conducted an interesting study, however, with a poor methodological foundation. In short, why was there no control group?  

The abstract does not present results with p values. Correct

Introduction

The introduction is, in general solid. However, you have written in lines 59 and 60 ‘’ The implementation of experimental forms of modern training in kickboxing has not been described in the literature to date’’ So from the next paragraph, I presume CrossFit is one of these modern training. However, this statement is false. Here are 2 papers that have used Cross fit in kickboxing; https://doi.org/10.1016/j.jbmt.2020.11.016 and  10.3233/IES-203190. This is also connected to the next paragraph, where you should report studies that have already used CrossFit in kickboxing and their effect. Therefore, please thoroughly review the scientific literature again and amend the paragraphs accordingly!

Also, there is no clear rationale for selecting the tests in testing battery.

Methods

Report how did you determine your sample size – by G*Power or any other methods?

What was the gender of your sample? report

Line 75 – define what makes them elite (categorisation, medal, etc.) as from your sample description, it is hard to convince me that someone with 3 years of training experience (lower part of your SD) can be considered an elite athlete. Also, training and competition experience are not the same. Please add competition time experience.

Define exactly what does it mean ‘’ minimum class 1 athletes’’.

Define better exclusion criteria. What about the status of injuries?

How can you report morphological measurements in the ‘’study group section’’? First, create a separate subheading and report it as it should be reported in detail – conditions athletes need to meet before the measurements to be accurate. What was the time of the measurements?

I which part of the season were your participants – pre-season, competitive, etc. report

How were the participants divided into two groups? Report

Experimental training:

Please provide supplemental material with photo material of the training.

Additionally, report the height of the box for jumping. Was it the same height for everyone or adapted to the body height? Report

Where was the training executed, at what part of the training and who supervised the training? Report

Physical fitness testing

For all tests, report how many times each test was performed, the sequence of tests, and the break between the tests. Also, report ALL equipment used, manufacturer and model! Be consistent and describe each test to the same amount of detail as others.

  1. Handgrip – connect the first two sentences in the test description. What was the break between the repetitions? How did you determine the dominant hand? What is the exact model of the dynamometer (I can’t believe you wrote ‘’made in USSR’’)? Be exact in the body position and report shoulder and elbow angle. Why just 2 reps – back up by references and why max result and not average?
  2. What was the height of a gym bench? What was the starting position?

Lines 158-159 ‘'A tape measure, a hard surface, and two gymnastic mattresses connected lengthwise’’ What is the meaning of this sentence? Why were the mattresses used for?

  1. Poor description – rewrite and add all necessary info that a test needs for a scientific paper!

Results.

The body composition data are missing

Discussion is unstructured and hard to follow. You are just commenting on test results but not putting it in the context of training and other studies and other training methods. This is the core of your paper and not tests!  The discussion is mainly missing the point.

You wrote ‘’Therefore, significant improvements in this ability cannot be expected’’ really? First, to whom did you compare the data? Second, why didn’t you report vo2 max values for the cooper test – it can easily be recalculated? What about some critical thinking that perhaps another test would yield more accurate results. Therefore, your conclusions are insignificant and should not be put in this way!

No limitations of the study paragraph    

What about the practical applications?

Conclusion – No comment as it is hard to believe this is your conclusion. I hope you can figure out what is missing.

Overall an interesting topic; however, a really poorly written paper with poor methodology and no control group. Therefore, I am rejecting this paper.

Kind regards

Author Response

Dear Reviewer,

Thank you very much for your time and valuable comments, which all have been considered and incorporated. The detailed list of responses is given below. We hope that the modifications and explanation will be acceptable for you.

Yours sincerely,

Rydzik, corresponding author

Dear Authors

You have conducted an interesting study, however, with a poor methodological foundation. In short, why was there no control group?  

A; The control group was not presented for clarity of statistical analysis. However, after your comment, we decided to correct it by including a control group.

The abstract does not present results with p values. Correct

A: This part has been corrected

Introduction

The introduction is, in general solid. However, you have written in lines 59 and 60 ‘’ The implementation of experimental forms of modern training in kickboxing has not been described in the literature to date’’ So from the next paragraph, I presume CrossFit is one of these modern training. However, this statement is false. Here are 2 papers that have used Cross fit in kickboxing; https://doi.org/10.1016/j.jbmt.2020.11.016 and  10.3233/IES-203190. This is also connected to the next paragraph, where you should report studies that have already used CrossFit in kickboxing and their effect. Therefore, please thoroughly review the scientific literature again and amend the paragraphs accordingly!

A: Thank you for your valuable comments. The introduction has been improved.

Also, there is no clear rationale for selecting the tests in testing battery.

A: The rationale has been added below the tests in the Methods section.

Methods

Report how did you determine your sample size – by G*Power or any other methods?

A: Information has been added

What was the gender of your sample? Report

A: Information has been added

Line 75 – define what makes them elite (categorisation, medal, etc.) as from your sample description, it is hard to convince me that someone with 3 years of training experience (lower part of your SD) can be considered an elite athlete. Also, training and competition experience are not the same. Please add competition time experience.

A: This part has been corrected

Define exactly what does it mean ‘’ minimum class 1 athletes’’.

A: In Poland, athletes are classified by sports skill classes (categories). There are classes 1, 2, and 3, with each class determined based on winning during competitions.

Define better exclusion criteria. What about the status of injuries?

A: This part has been expanded

How can you report morphological measurements in the ‘’study group section’’? First, create a separate subheading and report it as it should be reported in detail – conditions athletes need to meet before the measurements to be accurate. What was the time of the measurements?

A: Relevant information has been added

I which part of the season were your participants – pre-season, competitive, etc. Report

A: During the preparation period; information has been added.

How were the participants divided into two groups? Report

A: Information has been added

Experimental training:

Please provide supplemental material with photo material of the training.

A: Unfortunately, we do not have any photo documentation of the training. Furthermore, this would mean the publication of the images without the consent of the respondents.

Additionally, report the height of the box for jumping. Was it the same height for everyone or adapted to the body height? Report

A: A standard 40cm high CrossFit box was used for all participants.

Where was the training executed, at what part of the training and who supervised the training? Report

Training took place in the kickboxing club Centrum Aktywności Fizycznej in Rzeszów, Poland during the preparation period. The training was supervised by two kickboxing master class coaches.

Physical fitness testing

For all tests, report how many times each test was performed, the sequence of tests, and the break between the tests. Also, report ALL equipment used, manufacturer and model! Be consistent and describe each test to the same amount of detail as others.

A: Information has been added

  1. Handgrip – connect the first two sentences in the test description. What was the break between the repetitions? How did you determine the dominant hand? What is the exact model of the dynamometer (I can’t believe you wrote ‘’made in USSR’’)? Be exact in the body position and report shoulder and elbow angle. Why just 2 reps – back up by references and why max result and not average?

A: The sentences have been connected, while handedness is included in the test description. The test was performed with the right and left hand while the better result was recorded. The choice of the best score resulted from the test description (Grabowski H, 1989; Talaga, 2004). We apologize for our mistake regarding the manufacturer, the correct information has been provided. The mistake resulted from copying the test description. 

  1. What was the height of a gym bench? What was the starting position?

A: Information has been added

Lines 158-159 ‘'A tape measure, a hard surface, and two gymnastic mattresses connected lengthwise’’ What is the meaning of this sentence? Why were the mattresses used for?

Thank you for the valuable comment. This part has been removed.

  1. Poor description – rewrite and add all necessary info that a test needs for a scientific paper!

Results.

The body composition data are missing

A: By rewriting the manuscript, we have removed the description of body composition and presented it only in the group description.

Discussion is unstructured and hard to follow. You are just commenting on test results but not putting it in the context of training and other studies and other training methods. This is the core of your paper and not tests!  The discussion is mainly missing the point.

A: The discussion has been corrected

You wrote ‘’Therefore, significant improvements in this ability cannot be expected’’ really? First, to whom did you compare the data? Second, why didn’t you report vo2 max values for the cooper test – it can easily be recalculated? What about some critical thinking that perhaps another test would yield more accurate results. Therefore, your conclusions are insignificant and should not be put in this way!

A: This has been corrected as suggested. VO2max values were not reported because we think they are inaccurate based on the Cooper test.

No limitations of the study paragraph    

A: Dodano limitation

What about the practical applications?

A: Relevant information has been added

Conclusion – No comment as it is hard to believe this is your conclusion. I hope you can figure out what is missing.

A: This part has been corrected

Overall an interesting topic; however, a really poorly written paper with poor methodology and no control group. Therefore, I am rejecting this paper.

A: Thank you for your critical review, the paper has been thoroughly rewritten. We hope that we have met your expectations. Thank you

Reviewer 3 Report

The authors have conducted a study to test the effect of crossfit in kickboxers. There are many unclear statements all over the manuscript and lack of focus. Several grammatical errors. Introduction is not strong enough to build the case of as to why the study was done which questions the novelty of the study. Please see the attached file for my comments.

Author Response

Dear Reviewer,

Thank you very much for your time and valuable comments, which all have been considered and incorporated. All your comments have been taken into account and included in the manuscriptw. We hope that the modifications and explanation will be acceptable for you.

Yours sincerely,

Rydzik, corresponding author

  1. What was the intervention here?

A: Added information in abstract  line 22-23

  1. not sure what modern training means?

A: This has been correced, it was about implementing new taining forms. The whole intruduction was modified  line 72-78

  1. what modifications?

A: This has been corrected, it was a typing error because it was the author's training program line 83

  1. not clear if the study is to test the effect of crossfit training on kick-boxer performance or trying to design a program for athletes in general?

Introduction is all over place without focus. This must be rewritten to build a strong case as to what exactly has been done in the past, what's missing, why it is important, and what needs to be done then mention purpose of the study in those lines.

A: Dear reviewer , thank you for your comment, the introduction was rewritten and the purpose of the study was specified in detail. The study was designed to examine whether the author's crossfit-based training affects general and special fitness levels line  56-87

  1. define what it means.

A: This has been corrected „ Sample size was determined using G*Power software 3.1.9.4 [36] Participants were selected by purposive sampling i.e., subjective non-random sampling based on clear criteria for selecting a group for the study such as training experience, sports skill level (minimum class 1 athletes), and consent to participate in the experiment” line 91-107

  1. athletes? participants? subjects?

A: This has been corrected „athlets” line 102-103

  1. what is considered standard training and what modifications were made? was there a group with standard training and modified training? if not, how can the conclusion be made that modification have helped the athletes?

A: This represents the standard training process used in kickboxing. A control group was added to confirm the findings

  1. body mass? body fat percent or lean mass? also, cite PMID: 33472346 to show that this has been a validated tool for body composition in athletes.

A: Corrected and added citation. Thank you  line 110-114

  1. this table needs explanation. it is unclear what the authors are trying to communicate through this information

A: Added information below table  line 140-141

  1. define at first use

A: Added information below table  line 140-141

  1. why is everything in capital letters? and why not left aligned? it's rather difficult to read.

A: This has been corrected

  1. this information can be in the supplementary

A: Thank you for your comment however we feel it is more readable to post them here. In addition, a description of special ability tests has been added

  1. modified?

A: This has been corrected

Round 2

Reviewer 2 Report

Dear Authors,

Thank you for addressing some of my comments. However, some of them have not been addressed fully. Therefore further work needs to be done.

Introduction

The introduction is, in general solid. However, you have written in lines 59 and 60 ‘’ The implementation of experimental forms of modern training in kickboxing has not been described in the literature to date’’ So from the next paragraph, I presume CrossFit is one of these modern training. However, this statement is false. Here are 2 papers that have used Cross fit in kickboxing; https://doi.org/10.1016/j.jbmt.2020.11.016 and  10.3233/IES-203190. This is also connected to the next paragraph, where you should report studies that have already used CrossFit in kickboxing and their effect. Therefore, please thoroughly review the scientific literature again and amend the paragraphs accordingly!

A: Thank you for your valuable comments. The introduction has been improved.

Where have you added these two studies, as the references are still the same from the 1st version of the paper! So please add these two references in the introduction and in the references and update the whole references section. The in-text references count is 59, but the references are still at 46! This should already be done in this version and, unfortunately, just extends the review process from the author's side.

Also, there is no clear rationale for selecting the tests in testing battery.

A: The rationale has been added below the tests in the Methods section.

The rationale for the test selected should be presented in the introduction and NOT in the methods. Please amend the introduction accordingly.

Report how did you determine your sample size – by G*Power or any other methods?

A: Information has been added

Just adding – ‘’Sample size was determined using G*Power software 3.1.9.4’’ is not enough! Was it priori test, what was the power, tails, alpha error, effect size? Report

You have text in Polish!?? Please check this before submission of the manuscript! Correct

Define exactly what does it mean ‘’ minimum class 1 athletes’’.

A: In Poland, athletes are classified by sports skill classes (categories). There are classes 1, 2, and 3, with each class determined based on winning during competitions.

Add this to the text, as readers that don’t know the Polish sport categorisation system will appreciate this info. Amend accordingly.

Report body composition for control and experimental group. Add units in body fat. Report conditions athletes need to meet before the measurements, to be accurate. What was the time of the measurements?

Please provide supplemental material with photo material of the training.

A: Unfortunately, we do not have any photo documentation of the training. Furthermore, this would mean the publication of the images without the consent of the respondents.

You don’t need photos from the training. However, you can create new photos of the exercises for better repeatability of your study. Please take them and add this to the paper as supplemental material. Please add the material.

Physical fitness testing

For all tests, report how many times each test was performed, the sequence of tests, and the break between the tests. Also, report ALL equipment used, manufacturer and model! Be consistent and describe each test to the same amount of detail as others.

A: Information has been added

Not completely! Amend accordingly- Also, what are 10, 11 and 12 points empty?

The conclusion should not be written as 1), 2) and 3) ! Rewrite it

The practical application is not a separate subsection – Incorporate it in the conclusion.

Author Response

Dear Reviewer,

Thank you very much for your time and valuable comments, which all have been considered and incorporated. The detailed list of responses is given below. We hope that the modifications and explanation will be acceptable for you.

Yours sincerely,

Rydzik, corresponding author

Where have you added these two studies, as the references are still the same from the 1st version of the paper! So please add these two references in the introduction and in the references and update the whole references section. The in-text references count is 59, but the references are still at 46! This should already be done in this version and, unfortunately, just extends the review process from the author's side.

A: Sorry, we entered the publication via Mendeley and did not click on refresh, now everything is fine. You can find the publication under numbers 31 and 32.

The rationale for the test selected should be presented in the introduction and NOT in the methods. Please amend the introduction accordingly.

A: it has been corrected

Just adding – ‘’Sample size was determined using G*Power software 3.1.9.4’’ is not enough! Was it priori test, what was the power, tails, alpha error, effect size? Report

A: Yes, added . Detailed statistics added in the appendix at the end of the paper (Apenix A)

Report body composition for control and experimental group. Add units in body fat. Report conditions athletes need to meet before the measurements, to be accurate. What was the time of the measurements?

A: Dear reviewer. We would like to point out that the description given in section 2.4 applies to the entire study group, and the mean values are presented. Therefore, we do not provide a detailed write-up of the control and experimental groups. We only made the breakdowns later. A unit for body fat (%) has been added. The inclusion criteria were extended and the time of measurement was added

You don’t need photos from the training. However, you can create new photos of the exercises for better repeatability of your study. Please take them and add this to the paper as supplemental material. Please add the material.

A: A demo video has been added in the supporting material

Not completely! Amend accordingly- Also, what are 10, 11 and 12 points empty?

A: it has been corrected

The conclusion should not be written as 1), 2) and 3) ! Rewrite it

A: it has been corrected

The practical application is not a separate subsection – Incorporate it in the conclusion.

A: Of course you are right , it is not a separate chapter, but a subsection attached to the vnikniks. This entry is in line with the IJERPH standards. 

Reviewer 3 Report

The authors have well addressed my concerns. Reference 37 is inappropriate. The study did not validate BIA over DXA. Please consider citing previously recommended references or equivalent studies.

Author Response

Dear Reviewer,

Thank you very much for your time and valuable comments, which all have been considered and incorporated. The detailed list of responses is given below. We hope that the modifications and explanation will be acceptable for you.

Yours sincerely,

Rydzik, corresponding author

The authors have well addressed my concerns. Reference 37 is inappropriate. The study did not validate BIA over DXA. Please consider citing previously recommended references or equivalent studies.

A: This has been correct 

Round 3

Reviewer 2 Report

Dear Authors

Thank you for addressing the majority of raised comments. Some small portions should still be further addressed for greater consistency and clarity.

Video in the supplementary files is a great option. However, I see you probably used the trial version of the software as the watermark of the Movavi video editor plus is popping up every 20 and it is disturbing. Please remove it. Also, pay attention to editing as directly at the beginning, you have the template text showing up (add some text here), which is overlapping yours.

Additionally, the one video you created presents only the Monday workout. What about Wednesday and Friday? Please add

Also mention in the text that the detailed execution of exercises is available in the supplemental material.

2.2. Morphological characteristics of the subjects

I presume body composition and BODYWEIGHT was measured by Tanita BIA. Therefore please add that weight was also measured with it. Additionally, for bioimpedance analysis to be accurate, some predispositions need to be met, which you still did not report. Please state them. Usually, they are described similarly to this paper ( citation of the paper is left to authors discretion - this is just to show a direct example of what is meant by my comment): 10.3390/biology10111199  Page - 4 Experimental procedure paragraph - points from 1 to 7.

Also, please look at how you report G*Power measurements on page 3 of the mentioned paper, as what you have done is not sufficient.

Line 126 - you stated that the training was done 4 times per week. However, you report only 3 trainings for Monday, Wednesday and Friday. What was it then - 3 or 4 trainings per week? Amend accordingly

Handgrip measurement: Report the manufacturer of the handgrip dynamometer (MG 4800) (also check the grammar and spelling); what was the break between attempts; how did you determine the dominant hand? Report all these things

Be consistent as you report references in some of the tests (1 and 8) and not for the others (2-7 and 9). Please add references in every test you used.

Report heart rate monitor model and manufacturer used for the SKFT test.

And don’t forget to update the references at the end and please do English proofreading of the paper.

Overall I see a good improvement from the initial state. Therefore, I recommend acceptance after minor revision.

Kind regards

Author Response

Dear Reviewer,

Thank you very much for your time and valuable comments, which all have been considered and incorporated. The detailed list of responses is given below. We hope that the modifications and explanation will be acceptable for you.

Yours sincerely,

Rydzik, corresponding author

Video in the supplementary files is a great option. However, I see you probably used the trial version of the software as the watermark of the Movavi video editor plus is popping up every 20 and it is disturbing. Please remove it. Also, pay attention to editing as directly at the beginning, you have the template text showing up (add some text here), which is overlapping yours.

A: Dear Reviewer, thank you for your detailed comments. The video has been reshot and expanded to include all the training days.

Additionally, the one video you created presents only the Monday workout. What about Wednesday and Friday? Please add

A: All training days have been added.

Also mention in the text that the detailed execution of exercises is available in the supplemental material.

A: The relevant information has been added under Table 2.

2.2. Morphological characteristics of the subjects

I presume body composition and BODYWEIGHT was measured by Tanita BIA. Therefore please add that weight was also measured with it. Additionally, for bioimpedance analysis to be accurate, some predispositions need to be met, which you still did not report. Please state them. Usually, they are described similarly to this paper ( citation of the paper is left to authors discretion - this is just to show a direct example of what is meant by my comment): 10.3390/biology10111199  Page - 4 Experimental procedure paragraph - points from 1 to 7.

A: Thank you for your help, we have complemented the information and procedure.

Also, please look at how you report G*Power measurements on page 3 of the mentioned paper, as what you have done is not sufficient.

A: I am sorry for my previous calculations. I did not fully understand your comment and calculated something completely different using the wrong tool. Again, I apologize for that. Sample size determined using G*Power software 3.1.9.4 showed that with the estimated moderate effect size, at least twenty participants were needed in each group (effect size f=0.65, power= 0.95, p= 0.05)[36]. In the present analysis, 30 participants in each group were recruited to ensure a more accurate analysis. We have added this information in the manuscript and hope it will be acceptable to you. 

Line 126 - you stated that the training was done 4 times per week. However, you report only 3 trainings for Monday, Wednesday and Friday. What was it then - 3 or 4 trainings per week? Amend accordingly

 A: Thank you for your comment, we have corrected this part: training was performed 3 times a week

Handgrip measurement: Report the manufacturer of the handgrip dynamometer (MG 4800) (also check the grammar and spelling); what was the break between attempts; how did you determine the dominant hand? Report all these things

A: Thank you for your comment, relevant information has been added.

Be consistent as you report references in some of the tests (1 and 8) and not for the others (2-7 and 9). Please add references in every test you used.

 A: Thank you, references have been added.

Report heart rate monitor model and manufacturer used for the SKFT test.

A: Thank you, we have reported the heart rate monitor model.

And don’t forget to update the references at the end and please do English proofreading of the paper.

 A: Thank you, we have updated bibliography and corrected English.

Overall I see a good improvement from the initial state. Therefore, I recommend acceptance after minor revision.

A: Thank you.